# High Precision Causal Model Evaluation with Conditional Randomization

**Chao Ma**
Microsoft Research
Cambridge, UK
chao.ma@microsoft.com

**Cheng Zhang**
Microsoft Research
Cambridge, UK
cheng.zhang@microsoft.com

## Abstract

The gold standard for causal model evaluation involves comparing model predictions with true effects estimated from randomized controlled trials (RCT). However, RCTs are not always feasible or ethical to perform. In contrast, conditionally randomized experiments based on inverse probability weighting (IPW) offer a more realistic approach but may suffer from high estimation variance. To tackle this challenge and enhance causal model evaluation in real-world conditional randomization settings, we introduce a novel low-variance estimator for causal error, dubbed as the pairs estimator. By applying the same IPW estimator to both the model and true experimental effects, our estimator effectively cancels out the variance due to IPW and achieves a smaller asymptotic variance. Empirical studies demonstrate the improved of our estimator, highlighting its potential on achieving near-RCT performance. Our method offers a simple yet powerful solution to evaluate causal inference models in conditional randomization settings without complicated modification of the IPW estimator itself, paving the way for more robust and reliable model assessments.

## 1 Introduction

**Experimental approaches for causal model evaluation**    Causal inference models aim to estimate the causal effects of treatments or interventions on outcomes of interest, given observational (sometimes experimental) data. A crucial step in developing and validating such models is to evaluate their prediction quality, that is, how well they can approximate the true treatment effects that would have been observed under different scenarios. A common approach for causal model evaluation is to launching a *new* randomized controlled trial (RCT) separately, which provides a reliable estimate of the true treatment effects by randomly assigning units to treatments. By comparing the RCT estimate with the prediction of the causal model, one can assess the *causal error* (Equation 1), a key metric to measure how well the model reflects the true effects.

Nevertheless, RCTs are not always accessible or preferable, as random manipulation of treatments might be impractical, ethically questionable, or excessively expensive. For instance, it may be impossible or inappropriate to randomly assign individuals to different lifestyles, environmental exposures, or social policies, as these may involve personal choices, preferences, or constraints that affect the willingness or ability to participate in the experiment. In such cases, causal machine learning researchers may resort to or augment their analyses with conditionally randomized designs, in which the allocation of participants into treatment and control groups is not purely random, but is instead based on a predetermined condition or factor.

Conditionally randomized experiments can vary in the level and unit of treatment assignment, such as individual, group, or setting. They also depend on different assumptions and methods to account for the potential confounding, selection, or measurement biases that arise from the lack of randomization.

37th Conference on Neural Information Processing Systems (NeurIPS 2023).

A common scenario is the non-random quantitative assignment paradigm [West et al., 2008], where a new set of treatment groups are selected given some quantitative covariates $X$, via an oracle model $P_{exp}(T|X)$, where $T$ is the treatment variable. For example, in sales strategy assignment, it is typical that resources are allocated towards products with higher revenue contributions, which introduces explicit confounding. A widely used method to estimate the true treatment effect in this setting is the inverse probability weighting (IPW) estimator [Rosenbaum and Rubin, 1983, Rosenbaum, 1987], which uses the propensity scores, the probability of receiving the treatment given the covariates, to weight the observed outcomes and adjust for the imbalance between the treatment groups. This IPW estimate can then be compared with the causal model prediction to evaluate the causal error.

**Limitations of current approaches** Despite the popularity of the IPW estimator for conditionally randomized experiments, it suffers from several limitations that affect its reliability for causal model evaluation. [Khan and Tamer, 2010] shows that IPW may have unbounded variance when the propensity scores are imbalanced. Moreover, the it may have poor finite sample performance and high sensitivity to the specification of the propensity score model [Busso et al., 2014]. Last but not the least, a surprising finding is that even the oracle IPW may be harmful for the estimation efficiency [Hahn, 1998, Hirano et al., 2003]. As a consequence, the causal error estimation based on the IPW estimator may not be accurate or robust, and may lead to misleading conclusions about the causal model quality. Existing work on addressing the variance issue of the IPW estimator often involves modifying the IPW estimator itself [Crump et al., 2009, Chaudhuri and Hill, 2014, Sasaki and Ura, 2022, Busso et al., 2014, Robins et al., 2007, Lunceford and Davidian, 2004, Imbens, 2004, Liao and Rohde, 2022]. However, these methods are mainly designed for treatment effect estimation rather than causal error estimation, and may introduce additional bias or complexity that may not be optimal for the purpose of causal model evaluation.

**Contributions overview** We focus on causal model evaluation with conditionally randomized trials, propose a novel method for low-variance estimation of causal error (Equation 1), and demonstrate its effectiveness over current approaches by achieving near-RCT performance. Our key insight is: to estimate the causal error, we can design a simple and effective low-variance estimation procedure without improving the IPW estimator for the true treatment effect. As shown in Figure 1, denote the ground truth effect as $\delta$ and the treatment effect of a causal model $\mathcal{M}$ as $\delta_{\mathcal{M}}$. Our goal to estimate the causal error $\hat{\Delta} := \hat{\delta}_{\mathcal{M}} - \hat{\delta}^{IPW}$ with low variance. Using conditionally randomized experiments, we have an estimator of the ground truth via IPW: $\hat{\delta}^{IPW}$. Instead of improving the IPW estimator, we replace the commonly used sample mean estimator $\hat{\delta}\mathcal{M}$ with its IPW counterpart, $\hat{\delta}_{\mathcal{M}}^{IPW}$, having the same realizations of treatment assignments as in $\hat{\delta}^{IPW}$. This allows the variance of $\hat{\delta}^{IPW}$ to be hedged by $\hat{\delta}^{IPW}\mathcal{M}$, reducing the variance of the causal error estimator $\hat{\Delta}$. Contrary to conventional estimation strategies, the pairs estimator, as we call it, effectively reduces the variance of the causal error estimation and provides more reliable evaluations of causal model quality, both theoretically and empirically.

The rest of this paper is structured as follows. In Section 3, we formally describe our problem setting, necessary background and notations. In Section 4, we will formally define the pairs estimator for causal model evaluation, and study its theoretical properties (Proposition 1) on variance reduction. Finally, in Section 5, we will conduct simulation studies and demonstrate the effectiveness of the proposed estimation approach, highlighting its potential on achieving near-RCT performance using only conditionally randomized experimental data.

## 2 Related works

**Alternative schemes to randomized controlled trials** There exists a number of different conditionally randomized trial schemes in the literature, for instance randomized encouragement designs [Angrist et al., 1996], non-random quantitative treatment assignment [Imbens and Rubin, 2015], and fully observational studies [Rubin, 2005]. In this work, we mainly consider the non-random quantitative treatment assignment setting, where the treatment groups are determined by some known function of covariates. Nevertheless, we have an emphasized focus on using conditionally randomized trials for evaluating the quality of any given causal inference models (trained on observational data), which is a fundamental setting for many practical causal model deployment procedure.

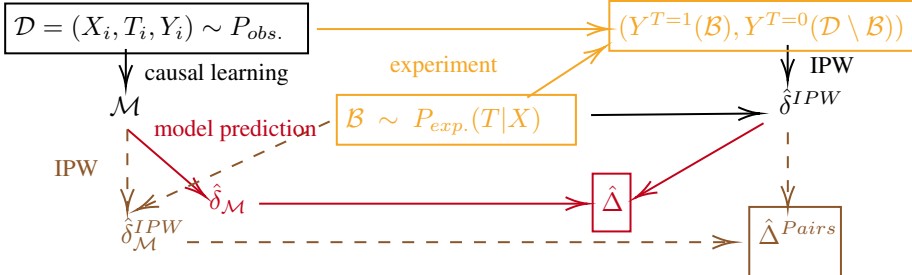

**Figure 1:** Overview of the proposed method. $\hat{\Delta}$: the naive estimator for causal error evaluation. $\hat{\Delta}^{Pairs}$: the proposed estimator. Given a causal model $\mathcal{M}$ learned from observational data, our goal is to estimate its causal error (Equation 1) via conditionally randomized experiments (yellow). Naive method (red) would achieve this by comparing $\hat{\delta}^{IPW}$, the IPW estimation of ground truth effect from the experiment, with $\hat{\delta}_{\mathcal{M}}$, the predicted treatment effect from $\mathcal{M}$, usually obtained via population mean (Equation 2). In our method (brown), we replace $\hat{\delta}_{\mathcal{M}}$ with its IPW counterpart, $\hat{\delta}_{\mathcal{M}}^{IPW}$, using the same treatment assignments used in estimating $\hat{\delta}^{IPW}$, so that the variance from IPW are hedged.

**Causal estimation methods with conditionally randomized experiments**    Apart from IPW mentioned in Section 1, a number of other methods were also developed for conditionally randomized trials settings, include matching [Ho et al., 2007, Stuart, 2010, Rubin, 2006], regression adjustment [Neyman, 1923, Rubin, 1974, Lin, 2013, Wager et al., 2016], double robust methods [Robins et al., 1995, Bang and Robins, 2005, Kang and Schafer, 2007, Athey et al., 2018], and machine learning methods [Athey and Imbens, 2016, Wager and Athey, 2018, Shi et al., 2019, Chernozhukov et al., 2018]. Readers can refer to [Imbens and Rubin, 2015, Morgan and Winship, 2015] for a comprehensive review. This paper, on the contrary, aims to propose a general method for causal model evaluation applicable to any causal effect estimation method, particularly the widely used IPW estimator.

**Variance reduction and robustness**    There has been a long standing discussion on robustness improvement [Robins et al., 1995] and variance reduction of IPW estimators. Several techniques have been proposed to address the problem of extreme or uninformative weights, including weights trimming [Crump et al., 2009, Chaudhuri and Hill, 2014, Sasaki and Ura, 2022], weights normalization [Busso et al., 2014, Robins et al., 2007, Lunceford and Davidian, 2004, Imbens, 2004], and more recently, linearly-modified IPW [Zhou and Jia, 2021], etc. These techniques aim to reduce variance but may introduce bias, complexity, or tuning parameters. Our work differs from the existing literature in that, we focus on the causal error estimation, rather than the individual treatment effect estimation. We do not improve the IPW estimator for the true treatment effect, but rather propose to apply the same IPW estimator to both the model and the true effects to hedge their variances.

## 3    Problem formulation

Consider the data generating distribution for the population on which the experiment is being carried over is given by $p(X, T, Y)$, where $X$ are some (multivariate) covariates, $Y$ is the outcome variable and $T$ is the treatment variable. In this paper, we only consider continuous effect outcomes. Let $Y^{T=t}$ denote the potential outcome of the effect variable under the intervention $T = t$. Without loss of generality, we assume $T \in \{0, 1\}$. Then, the interventional means are given by $\mu^1 = \mathbb{E}[Y^{T=1}]$, and $\mu^0 = \mathbb{E}[Y^{T=0}]$, respectively. The ground truth treatment effect is then given by

$$\delta = \mu^1 - \mu^0 = \mathbb{E}[Y^{T=1}] - \mathbb{E}[Y^{T=0}].$$

Now, assume that given observational data sampled from $p(X, T, Y)$, we have trained a causal model, denoted by $\mathcal{M}$, whose treatment effect is given by

$$\delta_{\mathcal{M}} = \mu_{\mathcal{M}}^1 - \mu_{\mathcal{M}}^0 = \mathbb{E}[Y_{\mathcal{M}}^{T=1}] - \mathbb{E}[Y_{\mathcal{M}}^{T=0}].$$

Our goal is then to estimate the *causal error* of the model, which quantifies how well the model reflects the true effects (the closer to zero, the better):

$$\Delta(\mathcal{M}) := \delta_{\mathcal{M}} - \delta \tag{1}$$

In practice, $\delta_{\mathcal{M}}$ will be the model output, and can be estimated easily. For example, we can sample a pool of i.i.d. subjects $\mathcal{D} = (X_1, Y_1), ..., (X_N, Y_N) \sim p(X, Y)$, and the corresponding treatment effect estimation will be given as

$$\delta_{\mathcal{M}} \approx \hat{\delta}_{\mathcal{M}} = \frac{1}{N} \sum_{i \in \mathcal{D}} [Y_{\mathcal{M}}^{T=1}(i) - Y_{\mathcal{M}}^{T=0}(i)], \tag{2}$$

which forms the basics of many casual inference methodologies, both for potential-outcome approaches and structural causal model approaches [Rubin, 1974, Rosenbaum and Rubin, 1983, Rubin, 2005, Pearl et al., 2000]. On the contrary, obtaining the ground truth effect $\delta$ is usually not possible without real-world experiments/interventions, due to the fundamental problem of causal inference [Imbens and Rubin, 2015]. By definition, $\delta$ can be (hypothetically) approximated by the population mean of potential outcomes:

$$\delta \approx \hat{\delta} := \frac{1}{N} \sum_{i \in \mathcal{D}} [Y^{T=1}(i) - Y^{T=0}(i)] \tag{3}$$

However, given a subject $i$, only one version of the potential outcomes can be observed. Therefore, we often resort to the experimental approach, the randomized controlled trial (RCT). The RCT approach is always considered as the golden standard for treatment effect estimation, in which we would randomly assign treatments to our pool of subjects $\mathcal{D} = (X_1, Y_1), ..., (X_N, Y_N) \sim p(X, Y)$, by flipping an unbiased coin. Then the estimated treatment effect is given by:

$$\hat{\delta}^{RCT} = \frac{1}{|\mathcal{B}|} \sum_{i \in \mathcal{B}} Y^{T=1}(i) - \frac{1}{N - |\mathcal{B}|} \sum_{j \in \mathcal{D} \setminus \mathcal{B}} Y^{T=0}(j)$$

where $\mathcal{B}$ denotes the subset of patients that are assigned with the treatment. Together, we have the *RCT estimator* of the causal error:

$$\hat{\Delta}^{RCT}(\mathcal{M}) := \hat{\delta}_{\mathcal{M}} - \hat{\delta}^{RCT} \tag{4}$$

However, when a randomized trial is not available, we can only deploy a conditionally randomized test assignment plan, represented by $\mathcal{T}$, which is a vector of $n$ Bernoulli random variables $\mathcal{T} = [b_1, ..., b_N]$, each determines that $Y_n^{T=1}$ will be revealed with probability $p_n$. In practice, $\mathcal{T}$ can be given by an *treatment assignment model* $p_{exp}(T = 1|X)$. This is an oracle distribution that is known and designed by the experiment designer. A subset of patients $\mathcal{B} \in \mathcal{D}$ is selected given these assigment probabilities. Then, the inverse probability weighted (IPW) estimation of the treatment effect is given by

$$\hat{\delta}^{IPW}(\mathcal{T}) := \frac{1}{N} < \mathbf{Y}^{T=1}(\mathcal{B}), \mathbf{w}(\mathcal{B}) > - \frac{1}{N} < \mathbf{Y}^{T=0}(\mathcal{D} \setminus \mathcal{B}), \frac{\mathbf{w}(\mathcal{D} \setminus \mathcal{B})}{\mathbf{w}(\mathcal{D} \setminus \mathcal{B}) - 1} >,$$

where $\mathbf{w} = [1/p_1, ..., 1/p_N]$, $\mathbf{w}(\mathcal{B})$ is created by sub-slicing $\mathbf{w}$ with subject indices in $\mathcal{B}$. $<,>$ is the inner product, and $\mathbf{1}$ is a vector of ones. Finally, the model causal error can be estimated as (dubbed *naive estimator* in this paper):

$$\hat{\Delta}(\mathcal{M}, \mathcal{T}) := \hat{\delta}_{\mathcal{M}} - \hat{\delta}^{IPW}(\mathcal{T}) \tag{5}$$

In practice, when the size $N$ of the subject pool is relatively small, the IPW estimated treatment effect $\hat{\delta}^{IPW}(\mathcal{T})$ will have high variance especially when $p_{exp}(T = 1|X)$ is skewed. As a result, one will expect a very high or even unbounded variance in the estimation [Khan and Tamer, 2010, Busso et al., 2014] $\hat{\Delta}(\mathcal{M}, \mathcal{T})$. The goal of this paper is then to improve model quality estimation strategy $\hat{\Delta}(\mathcal{M}, \mathcal{T})$, such that it has lower variance and error rates under conditionally randomized trials.

## 4 Pairs estimator for causal model quality evaluation

### 4.1 The pairs estimator

To resolve the problems with the naive estimator for causal error in Section 3, in this section, we propose a novel yet simple estimator that will significantly improve the quality of the causal error estimation in a model-agonist way. Intuitively, when estimating $\hat{\Delta}(\mathcal{M}, \mathcal{T})$, we can simply apply the

same IPW estimator (with the same treatment assignment) for *both* the model treatment effect $\delta_{\mathcal{M}}$ and the ground truth treatment effect $\delta$. In this way, we anticipate that the estimators for $\delta_{\mathcal{M}}$ and $\delta$ will become correlated; their estimation error will be canceled out and hence the overall variance is lowered. More formally, we have the following definition:

**Definition 1** (Pairs estimator for causal model quality). *Assume we have a pool of i.i.d. subjects to be tested, namely $\mathcal{D} = (X_1, Y_1), ..., (X_N, Y_N) \sim p(X, Y)$, as well as a conditionally randomized treatment assignment plan, represented by $\mathcal{T}$, which is a vector of $n$ Bernoulli random variables $\mathcal{T} = [b_1, ..., b_N]$, each determines that $Y_n^{T=1}$ will be revealed with probability $p_n$. Assume that, for a particular trial, a subset of patients $\mathcal{B} \in \mathcal{D}$ are selected using these probabilities. Then, the IPW estimator of the model's treatment effect and the ground truth treatment effect is given by*

$$\hat{\delta}_{\mathcal{M}}^{IPW}(\mathcal{T}) := \frac{1}{N} < \mathbf{Y}_{\mathcal{M}}^{T=1}(\mathcal{B}), \mathbf{w}(\mathcal{B}) > - \frac{1}{N} < \mathbf{Y}_{\mathcal{M}}^{T=0}(\mathcal{D} \setminus \mathcal{B}), \frac{\mathbf{w}(\mathcal{D} \setminus \mathcal{B})}{\mathbf{w}(\mathcal{D} \setminus \mathcal{B}) - 1} >$$

*, and*

$$\hat{\delta}^{IPW}(\mathcal{T}) := \frac{1}{N} < \mathbf{Y}^{T=1}(\mathcal{B}), \mathbf{w}(\mathcal{B}) > - \frac{1}{N} < \mathbf{Y}^{T=0}(\mathcal{D} \setminus \mathcal{B}), \frac{\mathbf{w}(\mathcal{D} \setminus \mathcal{B})}{\mathbf{w}(\mathcal{D} \setminus \mathcal{B}) - 1} >$$

*, respectively. Then, the pairs estimator of causal model quality is defined as*

$$\hat{\Delta}^{Pairs}(\mathcal{M}, \mathcal{T}) := \hat{\delta}_{\mathcal{M}}^{IPW}(\mathcal{T}) - \hat{\delta}^{IPW}(\mathcal{T}). \tag{6}$$

We will formally show that this new estimator can effectively reduce estimation error. We will provide the full assumptions and proposition below. Note that we will assume by default that the common assumptions for conditionally randomized experiments will hold, such as *non-interference/consistency/overlap*, etc, even though we will not explicitly mention them for the rest of the paper.

## 4.2 Core assumptions for achieving variance reduction

Our main assumption is regarding the estimation error of causal models, stated as below:

**Assumption A. Causal model estimation error for potential outcomes.** We assume that for each subject $i$, the trained causal model's potential outcome estimation can be expressed as

$$Y_{\mathcal{M}}^{T=t}(i) = Y^{T=t}(i) + V_i(Y^{T=t}(i)) * \nu_i, i = 1, 2, ..., N, t \in \{0, 1\} \tag{7}$$

where $\nu_i$ are i.i.d. distributed random error variables with *unknown* variance $\sigma_\nu^2$, that is independent from $Y_i^{T=t}$ and $b_i$; and $V_i(\cdot), i = 1, 2, ..., N$ is a set of deterministic function that is indexed by $i$. This assumption is very general and models the modulation effect between the independent noise $\nu$ and the ground truth counterfactual. One special example would be $Y_{\mathcal{M}}^{T=t}(i) = Y^{T=t}(i) + Y^{T=t}(i) * \nu_i$, where the estimation error will increase (on average) as $Y^{T=t}$ increases. In practice, dependencies between error magnitude and ground truth value could arise when the model is trained on observational data that suffers from selection bias, measurement error, omitted variable bias, etc. For the rest of paper, we write Equation 7 in its vectorized form:

$$\mathbf{Y}_{\mathcal{M}}^{T=t} = \mathbf{Y}^{T=t} + \mathbf{V}(\mathbf{Y}^{T=t}) * \boldsymbol{\nu},$$

where all operations are point-wise. Finally, we further require that the causal model's counterfactual prediction should be somewhat reasonable, in the sense that $\sigma_\nu^2 \mathbb{E}[(V_i Y^{T=t}(i))^2] < \mathbb{E}[Y^{T=t}(i)^2], t = 1, 0; i = 1, 2, ..., N$. This implies that the variance of the estimation error should at least be smaller than the ground truth counterfactual.

**Assumption B. Asymptotic normality of mean estimators of potential outcome variables.** [1] Let $\overline{Y^{T=1}}, \overline{Y^{T=0}}, \overline{Y_{\mathcal{M}}^{T=1}}, \overline{Y_{\mathcal{M}}^{T=0}}$ be the corresponding mean estimator of the potential outcome variables $Y^{T=1}, Y^{T=0}, Y_{\mathcal{M}}^{T=1}, Y_{\mathcal{M}}^{T=0}$. We assume that these estimators are jointly asymptotic normal, i.e.,

$$\sqrt{N}(\overline{Y^{T=1}} - \mathbb{E}Y^{T=1}, \overline{Y^{T=0}} - \mathbb{E}Y^{T=0}, \overline{Y_{\mathcal{M}}^{T=1}} - \mathbb{E}Y_{\mathcal{M}}^{T=1}, \overline{Y_{\mathcal{M}}^{T=0}} - \mathbb{E}Y_{\mathcal{M}}^{T=0})$$

converge in distribution to a zero mean multivariate Gaussian. This is reasonable due to the randomization and the large samples used in experiments [Casella and Berger, 2021, Deng et al., 2018].

---

[1]Note that this particular assumption is mainly for achieving asymptotic normality of the estimators, which is orthogonal to achieving variance reduction effect of our pairs estimator, that relies on **Assumption A**.

## 4.3 Theoretical results

Following our assumptions, we can derive the following theoretical result, which shows that given our assumptions described in the previous section, the pairs estimator $\hat{\Delta}^{Pairs}(\mathcal{M}, \mathcal{T})$ will effectively reduce estimation variance compared to the naive estimator, $\hat{\Delta}(\mathcal{M}, \mathcal{T})$.

**Proposition 1** (Variance reduction effect of the pairs estimator). *With the assumptions stated in Section 4.2, we can show that our IPW estimators, $\hat{\delta}^{IPW}(\mathcal{T})$ and $\hat{\delta}_{\mathcal{M}}^{IPW}(\mathcal{T})$ can be decomposed as*

$$\hat{\delta}^{IPW}(\mathcal{T}) = \hat{\delta} + f(\mathcal{B}), \quad \hat{\delta}_{\mathcal{M}}^{IPW}(\mathcal{T}) = \hat{\delta}_{\mathcal{M}} + f(\mathcal{B}) + g(\boldsymbol{\nu}, \mathcal{B}),$$

*where $f$ and $g$ are random variables that depend on $\mathcal{B}$ (or also $\boldsymbol{\nu}$), and $g(\boldsymbol{\nu}, \mathcal{B})$ is orthogonal to $\hat{\delta}$, $\hat{\delta}_{\mathcal{M}}$ and $f(\mathcal{B})$. Furthermore, if we define the estimation error of model quality estimators as follows*

$$e(\hat{\Delta}^{Pairs}(\mathcal{M}, \mathcal{T})) := \hat{\Delta}^{Pairs}(\mathcal{M}, \mathcal{T}) - \Delta(\mathcal{M})$$

$$e(\hat{\Delta}(\mathcal{M}, \mathcal{T})) := \hat{\Delta}(\mathcal{M}, \mathcal{T}) - \Delta(\mathcal{M}),$$

*then both $\sqrt{N}e(\hat{\Delta}^{Pairs}(\mathcal{M}, \mathcal{T}))$ and $\sqrt{N}e(\hat{\Delta}(\mathcal{M}, \mathcal{T}))$ are asymptotically normal with zero means, and their variances satisfies*

$$\mathbb{V}ar[e(\hat{\Delta}^{Pairs}(\mathcal{M}, \mathcal{T}))] < \mathbb{V}ar[e(\hat{\Delta}(\mathcal{M}, \mathcal{T}))].$$

Proof: See Appendix A ☐

This result provides theoretical justifications that our simple estimator will be effective for variance reduction. In the next section, we will conduct simulation studies to empirically evaluate and validate the effectiveness proposed approach.

# 5 Simulation Studies

In this section, we will evaluate the performance of the proposed pairs estimator, and validate our theoretical insights via simulation studies. We will also examine the robustness and sensitivity of the pairs estimator concerning different scenarios of conditionally randomized trials, including treatment assignment mechanisms, degree of imbalance, choice of causal machine learning models, etc.

## 5.1 Synthetic csuite dataset with hypothetical causal model

**Setting.** Following Geffner et al. [2022], we construct a set of synthetic datasets, designed specifically for evaluating causal inference performance, namely the csuite datasets. The data-generating process is based on structural causal models (SCMs), different levels of confounding, heterogeneity, and noise types are incorporated, by varying the strength, direction, and parameterize of the causal effects. We evaluated on three different datasets, namely csuite_1, csuite_2, and csuite_3, each with a different SCM. See Appendix B for more details. The corresponding causal model estimation is simulated using a special form of **Assumption A**, that is:

$$Y_{\mathcal{M}}^{T=t}(i) = Y^{T=t}(i) + Y^{T=t}(i) * \nu_i, i = 1, ..., N, t \in \{0, 1\}$$

where $\nu_i$ are i.i.d. distributed zero-mean random variables with variance $\sigma_{\nu}^2$ that affects the ground truth causal error. To simulate the conditionally randomized trials, we use two different schemes to generate the treatment assignment plans $\mathcal{T}$. The first scheme is based on a *logistic regression model* of the treatment assignment probability given the covariates, that is,

$$p_{exp}(T = 1|X) = \frac{1}{1 + \exp(-\beta^T X)}$$

, where $\beta$ is a random vector sampled from multivariate Gaussian distributions with mean zero and variance $\sigma_{\beta}^2$. We vary the degree of imbalance in the treatment assignment by changing the value of $\sigma_{\beta}^2$. A larger $\sigma_{\beta}^2$ implies a more imbalanced treatment assignment, as the variance of the treatment assignment probability increases. The second scheme is based on a *random subsampling* of the units, where the treatment assignment probability is fixed for each unit, but different units are sampled with replacement to form different treatment assignment plans. We vary the number of treatment assignment plans by changing the sample size of each subsample.

**Evaluation method.** We compare the performance of the pairs estimator with the naive estimator (Equation 5), as well as the RCT estimator (Equation 4), which is the ideal benchmark for causal model evaluation. We also compare with two other baselines, by replacing the IPW component $\hat{\delta}^{IPW}(\mathcal{T})$ in the naive estimator (Equation 5) by its variance reduction variants. This includes the self-normalized estimator, as well as the linearly modified (LM) IPW estimator Zhou and Jia [2021], a state-of-the-art method for IPW variance reduction when the propensity score is *known*. We measure the performance of the estimators by the following metrics: the variance, the bias, and the MSE of the causal error estimation. See Appendix B.1 for detailed definitions. We compute these metrics by averaging over 100 different realizations of the treatment assignment plans for each dataset.

**Results.** The results are shown in Figure 2 and Figure 3. Figure 2 shows the results for the logistic regression scheme, with different values of $\sigma_\nu^2$ and $\sigma_\beta^2$. Figure 3 shows the results for the random subsampling scheme, with different $\sigma_\nu^2$. In both tables, we report the average and the standard deviation of the performance metrics of the estimators for each value of the true causal error, which is computed by the difference between the true treatment effect and the model treatment effect.

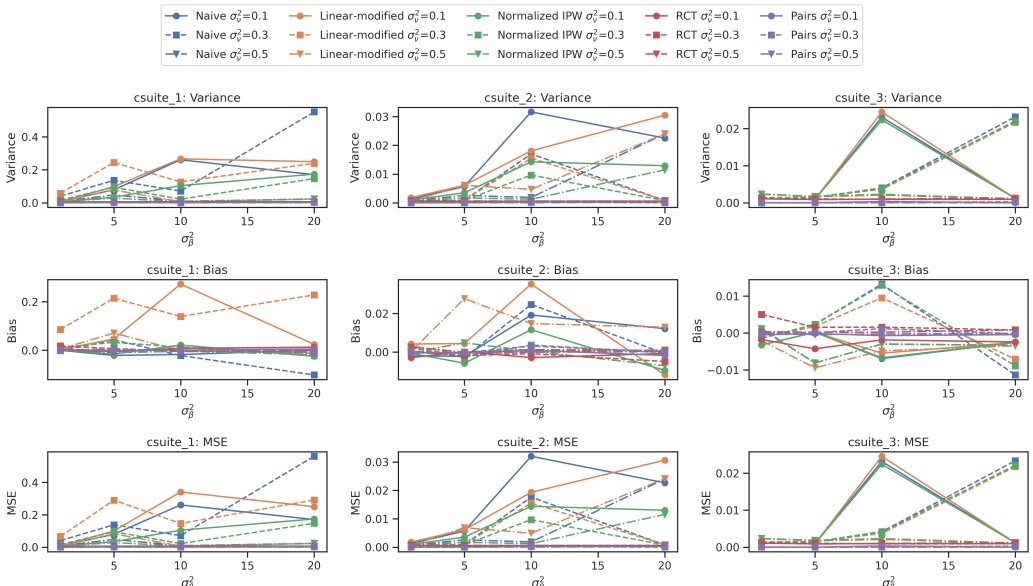

**Figure 2:** Comparison of various causal error estimators for the logistic assignment scheme across (`csuite_1`, `csuite_2`, and `csuite_3`). In this $3 \times 3$ plot, each row corresponds to a specific performance metric (Variance/Bias/MSE), while each column represents different datasets. In each plot, the x-axis represents the degree of treatment assignment imbalance ($\sigma_\beta^2$), while the y-axis displays the performance metrics (lower the better). The different colors indicate the performance of different estimators, and different linestyles and markers indicate different $\sigma_\nu^2$ settings. The results demonstrate that the proposed estimator (purple) consistently outperforms the other estimators in terms of all performance metrics, with a more robust behavior as the imbalance in treatment assignment increases.

From the tables, we can see that the variance of the naive estimator quickly increases when the treatment assignment is highly imbalanced. Nevertheless, our estimator (purple) consistently outperforms the naive estimator and its variance reduction variants in all metrics (variance/bias/MSE) regardless of the value of ($\sigma_\nu^2$), and the degree of imbalance ($\sigma_\beta^2$). The pairs estimator also achieves comparable performance to the RCT estimator, the golden standard by design. The linearly-modified estimators and self-normalized estimators have a lower variance than the naive estimator, but it also introduces some bias, and their performance is sensitive to the degree of imbalance. These demonstrate the effectiveness and robustness of the proposed pairs estimator when all assumptions are met.

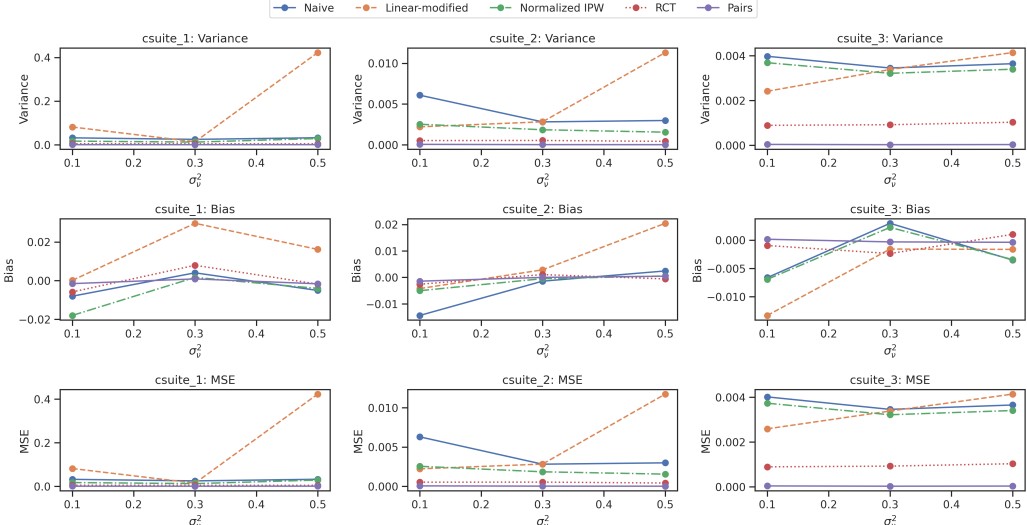

**Figure 3:** Comparative analysis of multiple causal error estimators in the context of the random subsampling scheme for (`csuite_1`, `csuite_2`, and `csuite_3`). In each plot, the x-axis represents the variance of the noise variable ($\sigma_\nu^2$), and the y-axis illustrates the performance metrics (Variance/Bias/MSE) for each estimator. Different colors denote the performance of different estimators. The findings reveal that, under the random subsampling scheme, the proposed estimator (purple) also consistently surpasses other estimators in every performance metric.

## 5.2 Synthetic counterfactual dataset with popular causal inference models

**Settings.** Based on Section 5.1, we now consider more realistic experimental settings, in which we apply a wide range of machine learning-based causal inference methods by training them from synthetic observational data. Thus, we expect the assumptions in Section 4.2 might not strictly hold anymore, which can be used to test the robustness of our method. We include a wide range of methods, such as linear double machine learning[Chernozhukov et al., 2018] (dubbed as `DML Linear`), kernel DML (`DML Kernel`) [Nie and Wager, 2021], causal random forest (`Causal Forest`) [Wager and Athey, 2018, Athey et al., 2019], linear doubly robust learning(`DR Linear`), forest doubly robust learning (`DR Forest`), orthogonal forest learning (`Ortho Forest`) [Oprescu et al., 2019], and doubly robust orthogonal forest (`DR Ortho Forest`). All methods are implemented via EconML package [Battocchi et al., 2019]. See Appendix B for more details.

We mainly focus on two aspects: 1), whether the proposed estimator can still be effective on variance reduction with non-hypothetical models as in Section 5.1; 2), whether the learned causal models' counterfactual predictions approximately follow the postulated **Assumption A**. Here We present the results for the first aspect, and results for the second can be found in Section 5.3.

**Simulation procedure.** We repeat the same simulation procedure as in Section 5.1, but using the learned causal inference models instead of the simulated causal model. We compare the performance of the pairs estimator with the same baselines as in Section 5.1, using the same metrics and the same treatment assignment schemes. For `DML Linear`, `DML Kernel`, `Causal Forest`, and `Ortho Forest` (that does not require propensity scores), we train on 2000 observational data points generated via the following data generating process of single continuous treatment [Battocchi et al., 2019]:

$$W \sim \text{Normal}(0,\ I^{n_w}), X \sim \text{Uniform}(0,1)^{n_x}$$
$$T = \langle W, \beta \rangle + \eta, \quad Y = T \cdot \theta(X) + \langle W, \gamma \rangle + \epsilon,$$

where $T$ is the treatment, $W$ is the confounder, $X$ is the control variable, $Y$ is the effect variable, and $\eta$ and $\epsilon$ are some uniform distributed noise. We choose the dimensionality of $X$ and $W$ to be $n_x = 30$ and $n_w = 30$, respectively. For other doubly robust-based methods, we use a discrete treatment that is sampled from a binary distribution $P(T = 1) = \text{sigmoid}(\langle W, \beta \rangle + \eta)$, while keeping the others unchanged. Once models are trained on the generated observational datasets, we use the

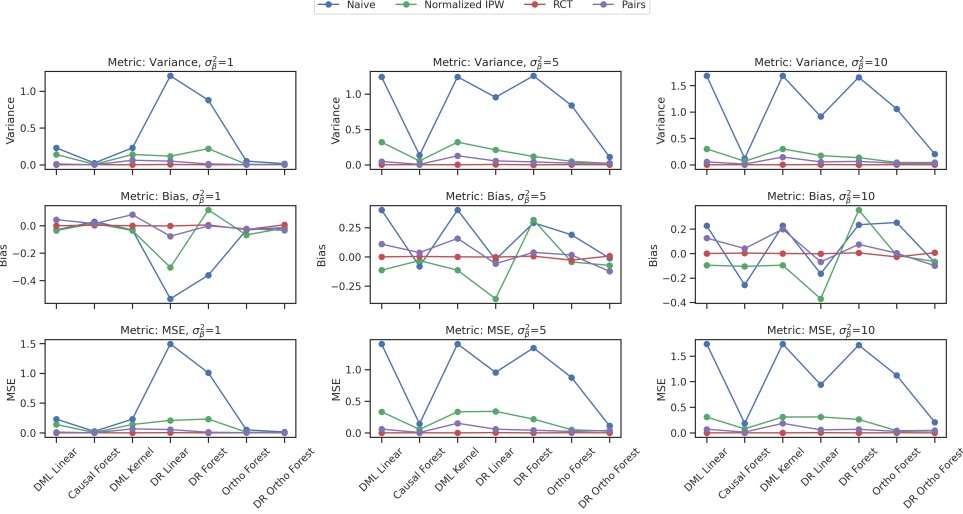

**Figure 4:** Performance of different causal error estimators across different causal inference methods. Each row of the grid corresponds to a specific performance metric (Variance/Bias/MSE), while each column represents different levels of treatment assignment imbalance ($\sigma_\beta^2$). In each plot, the x-axis represents different causal models. The y-axis displays performance metrics. Different colors correspond to different estimators. The results highlight the differences in performance among the estimators, with the pairs estimator (purple) consistently outperforming others in most scenarios, emphasizing its robustness and effectiveness in assessing various causal models.

trained causal inference model to estimate the potential outcomes and the treatment effects for each unit. Then, we use the logistic regression-based treatment assignment scheme as in Section 5.1 to simulate a hypothetical conditionally randomized experiment (for both continuous treatment and binary treatment, we will assign $T = 1$ for the treatment group and $T = 0$ for the control group). Both our pairs estimator as well as other baselines presented in Section 5.1 will be used to estimate the causal error. This is repeated 100 times across 3 different settings for treatment assignment imbalance ($\sigma_\beta^2 = 1, 5, 10$).

**Results.** Figure 4 [2] shows that using our estimator with conditionally randomized data, we achieved near-RCT performance for causal model evaluation quality across all metrics. It is also more robust to different settings of $\sigma_\beta^2$, where no significant variance change is observed. Also, we note that each causal error estimator has a different sweet-spot: for instance, the naive estimator usually consistently works very well with `Causal Forest`, whereas our estimator has relatively high variance and bias for `DML Kernel`. Nevertheless, the proposed estimator is still the most robust estimator (apart from RCT) that consistently achieves the best results across all causal models. This shows the feasibility of our method for reliable model evaluation with conditionally randomized experiments.

## 5.3 Validation of Assumption A in Section 4.2

**Settings** Finally, we provide experiments to validate Assumption A in Section 4.2. Specifically, we use a numerical example to check if Assumption A holds in practice for various commonly used models, including the ones considered in Section 5.2, including `DML Linear`, `DML kernel`, `Causal Forest`, and variations of doubly robust algorithms (`DR Linear`, `DR forest`, `DR Ortho Forest`). We first fit different methods using the observational data generated in Section 5.2, and obtain the counterfactual predictions from each method; Then, we parameterize the function $V(\cdot)$ in Equation 7 as a polynomial function and fit for the counterfactual predictions using Equation 7.

---

[2]The linearly-modified baseline is not displayed for clarity reasons, see Appendix B, Figure 6 for full results.

Finally, we solve for the residuals $\hat{\nu}$ in Equation 7, and test whether they are independent from $Y^T$ and $b$, as postulated in Assumption A.

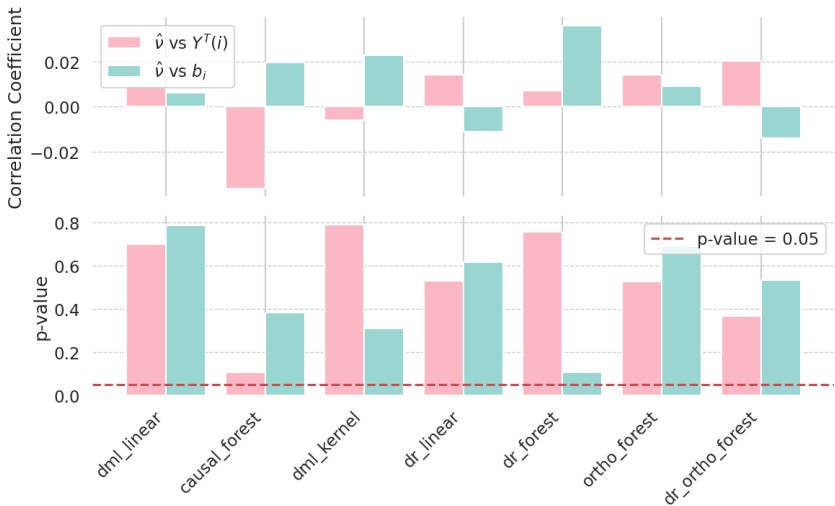

**Figure 5:** Validation of whether the learned causal models' counterfactual predictions approximately follow the postulated Assumption A.

**Results**    As shown in Figure 5, we calculate the Pearson correlation coefficient between $\hat{\nu}$ and $Y^T$ and $b$, respectively, as shown in the upper plot; we also compute the p-values for the null hypothesis that the samples are independent. We observe that the samples have close zero correlations as well as high p-values that are unable to reject the null assumption. Hence, both results shows suggests that the independence multiplicative noise assumption is likely to be true under this setting. We have also tried other tests such as Hoeffding's independence tests that captures non-linear dependencies, which yields similar results. This indicates that Assumption A is not overly restrictive and can be satisfied by a wide range of causal models that are popular in practical applications.

## 6    Conclusions

In this paper, we proposed the pairs estimator, a novel methodology for low-variance estimation of causal error in conditionally randomized trials. This approach applies the same IPW method to both the model and ground truth effects, canceling out the variance due to IPW. Remarkably, the pairs estimator can achieve near-RCT performance using conditionally randomized experiments, signifying a novel contribution for enabling more reliable and accessible model evaluation, without depending on expensive or infeasible randomized experiments. Future work may extend our method to more complex scenarios, explore alternative ways to reduce causal error estimation variance, and apply our method to other causality applications such as policy evaluation, causal discovery, and counterfactual analysis.

## Acknowledgements

We thank Colleen Tyler, Maria Defante, and Lisa Parks for conversations on real-world use cases that inspired this work. We also thank Nick Pawloski, Wenbo Gong, Joe Jennings, Agrin Hilmkil and Meyer Scetbon for useful discussions and support that enables this work.

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
