# A Proof of Proposition 1

Proof:

First, it is straightforward to show that the IPW estimator of the ground truth treatment effect $\hat{\delta}^{IPW}(\mathcal{T})$ can be re-written in terms of the population mean estimator, $\hat{\delta}$ (Equation 3):

$$\hat{\delta}^{IPW}(\mathcal{T}) = \hat{\delta} + \frac{1}{N} < \mathbf{Y}^{T=1}(\mathcal{B}), \mathbf{w}(\mathcal{B}) - 1 > + \frac{1}{N} < \mathbf{Y}^{T=0}(\mathcal{B}), \mathbf{1} >$$
$$- \frac{1}{N} < \mathbf{Y}^{T=0}(\mathcal{D} \setminus \mathcal{B}), \frac{1}{\mathbf{w}(\mathcal{D} \setminus \mathcal{B}) - 1} > - \frac{1}{N} < \mathbf{Y}^{T=1}(\mathcal{D} \setminus \mathcal{B}), \mathbf{1} > \qquad (8)$$

Similarly, we can derive a similar relationship for the IPW model treatment effect estimator $\hat{\delta}_{\mathcal{M}}^{IPW}(\mathcal{T})$:

$$\hat{\delta}_{\mathcal{M}}^{IPW}(\mathcal{T}) = \hat{\delta}_{\mathcal{M}} + \frac{1}{N} < \mathbf{Y}_{\mathcal{M}}^{T=1}(\mathcal{B}), \mathbf{w}(\mathcal{B}) - 1 > + \frac{1}{N} < \mathbf{Y}_{\mathcal{M}}^{T=0}(\mathcal{B}), \mathbf{1} >$$
$$- \frac{1}{N} < \mathbf{Y}_{\mathcal{M}}^{T=0}(\mathcal{D} \setminus \mathcal{B}), \frac{1}{\mathbf{w}(\mathcal{D} \setminus \mathcal{B}) - 1} > - \frac{1}{N} < \mathbf{Y}_{\mathcal{M}}^{T=1}(\mathcal{D} \setminus \mathcal{B}), \mathbf{1} > \qquad (9)$$

By setting:

$$f(\mathcal{B}) = \frac{1}{N} < \mathbf{Y}^{T=1}(\mathcal{B}), \mathbf{w}(\mathcal{B}) - 1 > + \frac{1}{N} < \mathbf{Y}^{T=0}(\mathcal{B}), \mathbf{1} >$$
$$- \frac{1}{N} < \mathbf{Y}^{T=0}(\mathcal{D} \setminus \mathcal{B}), \frac{1}{\mathbf{w}(\mathcal{D} \setminus \mathcal{B}) - 1} > - \frac{1}{N} < \mathbf{Y}^{T=1}(\mathcal{D} \setminus \mathcal{B}), \mathbf{1} >$$
$$g(\boldsymbol{\nu}, \mathcal{B}) = \frac{1}{N} < \boldsymbol{\nu}(\mathcal{B}) * \mathbf{V}(\mathbf{Y}^{T=1}(\mathcal{B})), \mathbf{w}(\mathcal{B}) - 1 > + \frac{1}{N} < \boldsymbol{\nu}(\mathcal{B}) * \mathbf{V}(\mathbf{Y}^{T=0}(\mathcal{B})), \mathbf{1} >$$
$$- \frac{1}{N} < \boldsymbol{\nu}(\mathcal{D} \setminus \mathcal{B}) * \mathbf{V}(\mathbf{Y}^{T=0}(\mathcal{D} \setminus \mathcal{B})), \frac{1}{\mathbf{w}(\mathcal{D} \setminus \mathcal{B}) - 1} > - \frac{1}{N} < \boldsymbol{\nu}(\mathcal{D} \setminus \mathcal{B}) * \mathbf{V}(\mathbf{Y}^{T=1}(\mathcal{D} \setminus \mathcal{B})), \mathbf{1} >$$

, Then under **Assumption A**, we arrive at the first conclusion of the proposition, that the estimation error of $\hat{\delta}^{IPW}(\mathcal{T})$ and $\hat{\delta}_{\mathcal{M}}^{IPW}(\mathcal{T})$ can be further decomposed as

$$\hat{\delta}^{IPW}(\mathcal{T}) = \hat{\delta} + f(\mathcal{B}) \qquad (10)$$
$$\hat{\delta}_{\mathcal{M}}^{IPW}(\mathcal{T}) = \hat{\delta}_{\mathcal{M}} + f(\mathcal{B}) + g(\boldsymbol{\nu}, \mathcal{B}) \qquad (11)$$

.

Therefore, $\hat{\Delta}^{Pairs}(\mathcal{M}, \mathcal{T})$ and $\hat{\Delta}(\mathcal{M}, \mathcal{T})$ are now given by

$$\hat{\Delta}^{Pairs}(\mathcal{M}, \mathcal{T}) = \hat{\delta}_{\mathcal{M}} - \hat{\delta} + g(\boldsymbol{\nu}, \mathcal{B}) \qquad (12)$$
$$\hat{\Delta}(\mathcal{M}, \mathcal{T}) = \hat{\delta}_{\mathcal{M}} - \hat{\delta} - f(\mathcal{B}) \qquad (13)$$

, respectively. Their estimation error is then given by

$$e(\hat{\Delta}^{Pairs}(\mathcal{M}, \mathcal{T})) = \hat{\delta}_{\mathcal{M}} - \hat{\delta} + g(\boldsymbol{\nu}, \mathcal{B}) - \Delta(\mathcal{M}) \qquad (14)$$
$$e(\hat{\Delta}(\mathcal{M}, \mathcal{T})) = \hat{\delta}_{\mathcal{M}} - \hat{\delta} - f(\mathcal{B}) - \Delta(\mathcal{M}) : \qquad (15)$$

. According to delta method Casella and Berger [2021], both $\sqrt{N}e(\hat{\Delta}^{Pairs}(\mathcal{M}, \mathcal{T}))$ and $\sqrt{N}e(\hat{\Delta}(\mathcal{M}, \mathcal{T}))$ are asymptotically normal with zero under **Assumption B**. However, their variances will differ. We proceed to compute the variances of each estimator.

First, note that $g(\boldsymbol{\nu}, \mathcal{B})$ can be re-written as

$$g(\boldsymbol{\nu}, \mathcal{B}) = \frac{1}{N} < \boldsymbol{\nu} * \mathbf{b} * \mathbf{V}(\mathbf{Y}^{T=1}), \mathbf{w} - 1 > + \frac{1}{N} < \boldsymbol{\nu} * \mathbf{b} * \mathbf{V}(\mathbf{Y}^{T=0}), \mathbf{1} >$$
$$- \frac{1}{N} < \boldsymbol{\nu} * (1 - \mathbf{b}) * \mathbf{V}(\mathbf{Y}^{T=0}), \frac{1}{\mathbf{w} - 1} > - \frac{1}{N} < \boldsymbol{\nu} * (1 - \mathbf{b}) * \mathbf{V}(\mathbf{Y}^{T=1}), \mathbf{1} > \qquad (16)$$

, where $b_i$ is the Bernoulli random variable with $P(b_i = 1) = p_i$, and $b_i = 1$ if $i \in \mathcal{B}$. Without loss of generality, here we additionally assume that $\nu$ has zero mean to further simplify the notational

complexity. The proof also holds for the non-zero mean case trivially. Therefore, note also that $\nu$ is independent from $(Y^T(i), b_i)$, we have

$$\mathbb{C}ov(Y^{T=t_a}(i), \nu_i b_i V_i(Y^{T=t_b}(i))) = \mathbb{E}(\nu_i b_i)\mathbb{C}ov(Y^{T=t_a}(i), V_i(Y^{T=t_b}(i)))$$
$$= 0$$

holds for all $i$ and all treatments $t_a$ and $t_b$. Similarly, we have $\mathbb{C}ov(Y_{\mathcal{M}}^{T=t_a}(i), \nu_i b_i V_i(Y^{T=t_b}(i))) = 0$.

Therefore, it is not hard to show that $\mathbb{C}ov(g(\boldsymbol{\nu}, \mathcal{B}), \hat{\delta}_{\mathcal{M}}) = 0$, and $\mathbb{C}ov(g(\boldsymbol{\nu}, \mathcal{B}), \hat{\delta}) = 0$, which implies $\mathbb{C}ov(g(\boldsymbol{\nu}, \mathcal{B}), \hat{\delta}_{\mathcal{M}} - \hat{\delta}) = 0$. Thus, we have:

$$\mathbb{V}ar[\sqrt{N}e(\hat{\Delta}^{Pairs}(\mathcal{M}, \mathcal{T}))] = \mathbb{V}ar[\sqrt{N}(\hat{\delta}_{\mathcal{M}} - \hat{\delta})] + \mathbb{V}ar[\sqrt{N}g(\boldsymbol{\nu}, \mathcal{B})]$$
$$= \mathbb{V}ar[Y_{\mathcal{M}}^{T=1} - Y_{\mathcal{M}}^{T=0} + Y^{T=1} - Y^{T=0}] + \mathbb{V}ar[\sqrt{N}g(\boldsymbol{\nu}, \mathcal{B})]$$
$$\mathbb{V}ar[\sqrt{N}e(\hat{\Delta}(\mathcal{M}, \mathcal{T}))] = \mathbb{V}ar[\sqrt{N}(\hat{\delta}_{\mathcal{M}} - \hat{\delta})] + \mathbb{V}ar[\sqrt{N}(f(\mathcal{B}))]$$
$$= \mathbb{V}ar[Y_{\mathcal{M}}^{T=1} - Y_{\mathcal{M}}^{T=0} + Y^{T=1} - Y^{T=0}] + \mathbb{V}ar[\sqrt{N}(f(\mathcal{B}))]$$

Since $b_i(1 - b_i) = 0$, we have

$$\mathbb{E}[\nu_i b_i V_i(Y_i^{T=t_a})\nu_i(1 - b_i)V_i(Y_i^{T=t_b})] = 0$$

. Therefore

$$\mathbb{V}ar[g(\boldsymbol{\nu}, \mathcal{B})] = \mathbb{V}ar[\frac{1}{N} < \boldsymbol{\nu} * \mathbf{b} * \mathbf{V}(\mathbf{Y}^{T=1}), \mathbf{w} - 1 >] + \mathbb{V}ar[\frac{1}{N} < \boldsymbol{\nu} * \mathbf{b} * \mathbf{V}(\mathbf{Y}^{T=0}), \mathbf{1} >]$$
$$+ \mathbb{V}ar[\frac{1}{N} < \boldsymbol{\nu} * (1 - \mathbf{b}) * \mathbf{V}(\mathbf{Y}^{T=0}), \frac{1}{\mathbf{w} - 1} >] + \mathbb{V}ar[\frac{1}{N} < \boldsymbol{\nu} * (1 - \mathbf{b}) * \mathbf{V}(\mathbf{Y}^{T=1}), \mathbf{1} >]$$

Since $\nu$ has zero mean and variance $\sigma_\nu^2$ and is independent of $(Y_i^T, b_i)$ as in **Assumption A**, this expression be further simplified as, according to the rules of variance of the product of independent variables:

$$\mathbb{V}ar[g(\boldsymbol{\nu}, \mathcal{B})] = \frac{1}{N^2} < \sigma_\nu^2 * \mathbf{p} * \mathbb{E}(\mathbf{V}(\mathbf{Y}^{T=1})^2), (\mathbf{w} - 1)^2 > + \frac{1}{N^2} < \sigma_\nu^2 * \mathbf{p} * \mathbb{E}(\mathbf{V}(\mathbf{Y}^{T=0})^2), \mathbf{1} >$$
$$+ \frac{1}{N^2} < \sigma_\nu^2 * (1 - \mathbf{p}) * \mathbb{E}(\mathbf{V}(\mathbf{Y}^{T=0})^2), \frac{1}{\mathbf{w} - 1} > + \frac{1}{N^2} < \sigma_\nu^2 * (1 - \mathbf{p}) * \mathbb{E}(\mathbf{V}(\mathbf{Y}^{T=1})^2), \mathbf{1} >$$
$$= \frac{\sigma_\nu^2}{N^2}[< \mathbf{p}(\mathbf{w} - 1)^2 + (\mathbf{1} - \mathbf{p}), \mathbb{E}(\mathbf{V}(\mathbf{Y}^{T=1})^2) > + < \frac{1 - \mathbf{p}}{\mathbf{w} - 1} + (\mathbf{p}), \mathbb{E}(\mathbf{V}(\mathbf{Y}^{T=0})^2) >]$$
$$< \frac{1}{N^2}[< \mathbf{p}(\mathbf{w} - 1)^2 + (\mathbf{1} - \mathbf{p}), \mathbb{E}((\mathbf{Y}^{T=1})^2) > + < \frac{1 - \mathbf{p}}{\mathbf{w} - 1} + (\mathbf{p}), \mathbb{E}((\mathbf{Y}^{T=0})^2) >]$$
$$= \mathbb{V}ar[f(\mathcal{B})]$$

Where the third equality is due to the fact that $\sigma_\nu^2\mathbb{E}[(V_i Y^{T=t}(i))^2] < \mathbb{E}[Y^{T=t}(i)^2]$. Therefore, we finally conclude that the variances of the error estimators will satisfy

$$\mathbb{V}ar[\sqrt{N}e(\hat{\Delta}^{Pairs}(\mathcal{M}, \mathcal{T}))] < \mathbb{V}ar[\sqrt{N}e(\hat{\Delta}(\mathcal{M}, \mathcal{T}))]$$

,

$\square$

# B  Additional details for experiment

## B.1  Implementation details

**Csuite dataset**   The csuite dataset used in Section 5.1 is an assortment of synthetic datasets first developed by [Geffner et al., 2022], for the purpose of evaluating both causal inference and discovery algorithms. They contain datasets ranging from small to medium scale (2-12 nodes), generated through carefully constructed Bayesian networks with additive noise models. All dataset in the collection includes a training set with 2,000 samples and 1 or 2 intervention or counterfactual test sets. The intervention test sets consist of factual variables, factual values, treatment variable, treatment value, reference treatment value, and an effect variable. More specifically, our three datasets corresponds to the following datasets:

1. `nonlin_simpson` (`csuite_1`): An example of Simpson's Paradox [Blyth, 1972] using a continuous SEM. The dataset is constructed so that $\text{Cov}(X_1, X_2)$ has the opposite sign to $\text{Cov}(X_1, X_2 \mid X_0)$. Estimating the treatment effects correctly in this SEM is highly sensitive to accurate causal discovery.

   The structural equations are

$$X_0 \sim N(0, 1) \tag{17}$$

$$X_1 = s(1 - X_0) + \sqrt{\frac{3}{20}} Z_1 \tag{18}$$

$$X_2 = \tanh(2X_1) + \frac{3}{2} X_0 - 1 + \tanh(Z_2) \tag{19}$$

$$X_3 = 5 \tanh\left(\frac{X_2 - 4}{5}\right) + 3 + \frac{1}{\sqrt{10}} Z_3 \tag{20}$$

   where $Z_1, Z_2 \sim N(0, 1)$ and $Z_3 \sim \text{Laplace}(1)$ are mutually independent and independent of $X_0$, $s(x) = \log(1 + \exp(x))$ is the softplus function. Constants were chosen so that each variable has a marginal variance of (approximately) 1.

2. `chain_lingauss` (`csuite_2`): Simulated from the graph $X_0 \to X_1 \to X)2$ with linear relationship. Ensure $X_0$, $X_1$ and $X_2$ have same standard deviation (1), then this turns into structural equations:

$$X_0 \sim \mathcal{N}(0, 1)$$

$$X_1 = \sqrt{\frac{2}{2}} X_0 + \sqrt{\frac{1}{3}} \mathcal{N}(0, 1)$$

$$X_2 = \sqrt{\frac{2}{3}} X_1 + \sqrt{(\frac{1}{3})} \mathcal{N}(0, 1)$$

3. `fork_lingauss` (`csuite_3`): Simulated from the graph $X_0 \leftarrow X_1 \to X_2$ with linear relationship. Turns into structural equations:

$$X_0 \sim \mathcal{N}(0, 1)$$

$$X_1 = \sqrt{\frac{2}{2}} X_0 + \sqrt{\frac{1}{3}} \mathcal{N}(0, 1)$$

$$X_2 = \sqrt{\frac{2}{3}} X_0 + \sqrt{\frac{1}{3}} \mathcal{N}(0, 1)$$

**Causal model details for Section 5.2**  In Section 5.2, We include a wide range of machine learning-based causal inference methods to evaluate the performance of causal error estimators. They can be roughly divided into 4 categories: double machine learning methods, doubly robust learning methods, ensemble causal methods, and orthogonal methods. All methods are implemented using `EconML` [Battocchi et al., 2019], as detailed below:

1. `DML Linear`: A linear double machine learning model [Chernozhukov et al., 2018], which uses an un-regularized final stage linear model for heterogenous treatment effect. Given that it is an unregularized low dimensional final model, this class also offers confidence intervals via asymptotic normality arguments. Random forests with default settings are used for first stage estimations.

2. `DML Kernel`: kernel DML with random Fourier feature approximations [Nie and Wager, 2021] and uns a ElasticNet regularized final model. Random forests with default settings are also used for first stage estimations. Others configs are kept as default.

3. `Causal Forest` causal random forest (or forest DML) [Wager and Athey, 2018, Athey et al., 2019]. We set the number of estimators to 100, number of minimum samples of leaf to 10, The number of samples to use for each subsample that is used to train each tree as 0.5. The others are kept as default. For effect and outcome models, we use Lasso with cross-validation.

4. `DR Linear` doubly robust learning with a final linear model. Regression model for $\mathbb{E}[Y|X, W, T]$ is set to random forest models. The propensity model is set to a logistic regression model.

5. `DR Forest` doubly robust learning with subsampled honest forest regressor. Regression model for $\mathbb{E}[Y|X, W, T]$ is set Gradient Boosting Regressor;and the propensity model is set to a random forest classifier. For other hyperparameters we set the minimum number of samples required to be at a leaf to be 10,and the minimum weighted fraction of the sum total of weights required to be at a leaf node to be 0.1.

6. `Ortho Forest`: orthogonal forest learning, a combination of causal forests and double machine learning that allow for controlling for a high-dimensional set of confounders, while at the same time estimating non-parametrically the heterogeneous treatment effect on a lower dimensional set of variables. We use Lasso with cross-validation as the estimator for residualizing both the treatment and the outcome at each leaf; and switch to weighted Lassos at prediction time. Readers may refer to the official documentation for more details, as well as discussions on the difference between this method and causal forest.

7. `DR Ortho Forest`: doubly robust orthogonal forest, a variant of the Orthogonal Random Forest that uses the doubly robust moments for estimation as opposed to the DML moments. Similarly, we use logistic regression models for residualizing the treatment at each leaf for both stages; and Lasso with cross-validation for the corresponding esimators for residualizing the outcomes. At prediction time, we switch to weighted Lasso instead.

**Evaluation metrics**   Throughout all experiments, We measure the performance of the estimators by the following metrics: the variance, the bias, and the MSE of the causal error estimation. More concretely, with a slight abuse of notation, let $\hat{\Delta}(\mathcal{M}, \mathcal{T}))$ denote the estimated causal error (from any estimation method). Then, the evaluation metrics are defined as:

$$Variance := \mathbb{E}_{\mathcal{T}}[\hat{\Delta}(\mathcal{M}, \mathcal{T}))^2] - \mathbb{E}_{\mathcal{T}}[\hat{\Delta}(\mathcal{M}, \mathcal{T}))]^2,$$
$$Bias := \mathbb{E}_{\mathcal{T}}[\hat{\Delta}(\mathcal{M}, \mathcal{T})) - \Delta(\mathcal{M})],$$
$$MSE := \mathbb{E}_{\mathcal{T}}[(\hat{\Delta}(\mathcal{M}, \mathcal{T})) - \Delta(\mathcal{M}))^2].$$

All expectations are taken over the treatment assignment plans $\mathcal{T}$. In practice, we draw 100 random realizations of treatment assignments and estimate all three metrics.

## B.2   Additional results

**Full results for causal error estimator metrics including linearly-modified IPW**

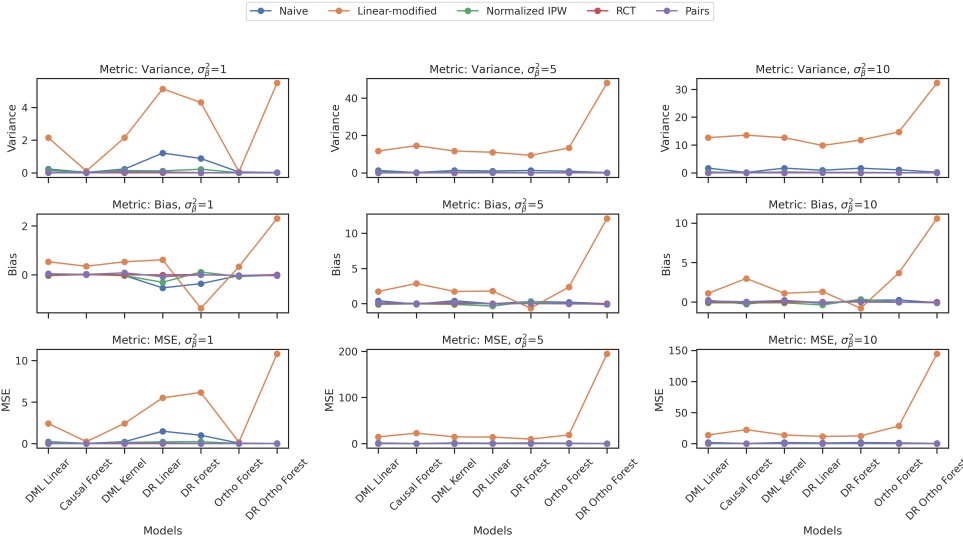

**Figure 6:** Causal error estimator quality metrics for across common causal methods (full results including linearly-modified IPW). Each row of the grid corresponds to a specific performance metric (Variance/Bias/MSE), while each column represents different levels of treatment assignment imbalance ($\sigma_\beta^2$). In each plot, the x-axis represents different causal models. The y-axis displays performance metrics. Different colors correspond to different estimators.