# OpenReview forum: "High Precision Causal Model Evaluation with Conditional Randomization"
_NeurIPS.cc/2023/Conference — NeurIPS 2023 poster_

### Official Review · Reviewer_yvJa · 2023-07-06

**Soundness:** 3 good
**Presentation:** 3 good
**Contribution:** 2 fair
**Rating:** 6
**Confidence:** 3

**Summary:**

The authors formulate and evaluate an approach to solving a non-standard problem: evaluating a causal model M when additional data (not used to construct M) is available from a non-randomized experiment. In particular, the authors focus on comparing IPW estimates from the non-RCT data and from the inferences of the model.

**Strengths:**

The idea is a simple and apparently powerful one: Remove the variability due to IPW by performing IPW on both the actual data and the estimates from the model. This largely removes an apparently extraneous source of variability (IPW itself) and allows direct comparison of the estimates.

**Weaknesses:**

The basic idea of the pairs estimator assumes some basic properties of the IPW estimator. Specifically, if IPW was a terrible estimator whose estimates were unrelated to the data (for example, it always output a single value for ATE: 0.5), then the pairs estimator would always show that the model was essentially perfect, regardless of the model’s estimates or the non-RCT data. I don’t think this scenario likely, but it is possible. This implies that, at least, some diagnostic tests are in order to increase confidence in the output of the pairs estimator. For example, you could introduce noise into the model’s estimates and see if the estimated error increases.

The results in Figures 2 and 3 are very good. Indeed, they are *freakishly* good. They are so good that it makes me wonder whether the experiments reported in these figures are really evaluating anything important about how the pairs estimator works in practice. This bears some discussion in the description of the results.

The introduction to the paper may be confusing to many readers. When first encountering the term “non-random experiment”, many readers will balk, thinking that randomization is the *sine qua non* of experimentation and seeing “non-random experiment” as a contradiction in terms. The authors could save readers this confusion by introducing the example of explicit non-random assignment (line 38) earlier or by moving the first paragraph of Section 2 (Related Works) to early in the introduction.

Another issue may confuse readers: The contrast between an IPW estimate and the “model’s” estimate of treatment effect. For many readers, the goal of analyzing observational data is to get a single estimate of treatment effect, perhaps through IPW. In this scenario, there is no “model” (or, the model is a model of treatment propensity). The authors could substantially improve the paper by explaining one or more practical scenarios in which a researcher has both a model of causal effect and a non-RCT data set that has not been used to create that model.

The paper has occasional small grammar errors that detract from the authors’ message. An example is the first sentence of the contributions (with corrections noted in brackets): “We focus on causal model evaluation with non-RCT[s], propose a novel method for low-variance estimation of causal error (Equation 1), and demonstrate its effective[ness] over current approaches by achieving near-RCT performance.” A little more care in editing would improve the paper.

**Questions:**

1. What are examples of practical scenarios in which researchers have a model of treatment effect and then they collect data non-RCT data to evaluate that model?

2. Under what circumstances would the pairs estimator fail to provide estimates with low variance and low bias?

**Limitations:**

The experiments do not identify cases in which the pairs estimator will fail. In the synthetic experiments whose results are shown in Figure 2 and 3 (which could be used to identify such failure modes), the results for the pairs estimator are freakishly good.

---

> ### Author Rebuttal · Authors · 2023-08-09
>
> Thank you very much for your review. Below we address your question respectively.
>
> > Some diagnostic tests are in order to increase confidence in the output of the pairs estimator. For example, you could introduce noise into the model’s estimates and see if the estimated error increases.
>
> Thank you for your suggestion, we will append failure scenarios of our method and try to introduce certain practical diagnoses.
>
>
> > The results in Figures 2 and 3 are very good. Indeed, they are freakishly good. They are so good that it makes me wonder whether the experiments reported in these figures are really evaluating anything important about how the pairs estimator works in practice. This bears some discussion in the description of the results.
>
> While the results in Figures 2 and 3 are indeed very good, they are based on the fact that the assumptions of our method are met or expected to be met. These results mainly validate the correctness of our theoretical results and demonstrate the practicality of our assumptions. In the revision, we will provide further discussion including potential failure cases and scenarios where the assumptions may not hold.
>
> > The introduction to the paper may be confusing to many readers. When first encountering the term “non-random experiment”, many readers will balk, thinking that randomization is the sine qua non of experimentation and seeing “non-random experiment” as a contradiction in terms. The authors could save readers this confusion by introducing the example of explicit non-random assignment (line 38) earlier or by moving the first paragraph of Section 2 (Related Works) to early in the introduction.
>
> Thank you so much for these suggestions. We recognize the naming conventions and may leads to confusion, which we originally followed [1].  In revision, we will apply your suggestions and mention different naming conventions in the literature.
>
> > What are examples of practical  scenarios in which researchers have a model of treatment effect and then they collect data non-RCT data to evaluate that model? The authors could substantially improve the paper by explaining one or more practical scenarios in which a researcher has both a model of causal effect and a non-RCT data set that has not been used to create that model.
>
> Such scenarios can be found in various applications. For example, in personalized medicine domain, researchers build complex models such as structural causal models (SCMs) to predict various types of causal quantities beyond ATE. In practice, performing trials on every single types of causal queries is expensive and ethically challenging, making the collection of non-RCT data a valuable downstream task to gain insights into the model's performance.
>
> > The paper has occasional small grammar errors that detract from the authors’ message. An example is the first sentence of the contributions (with corrections noted in brackets): “We focus on causal model evaluation with non-RCT[s], propose a novel method for low-variance estimation of causal error (Equation 1), and demonstrate its effective[ness] over current approaches by achieving near-RCT performance.” A little more care in editing would improve the paper.
>
> Thank you for your input. In revision, we will further polish and improve the paper.
>
>
> **References**
>
> [1] Rubin, Donald B. "Estimating causal effects of treatments in randomized and nonrandomized studies." Journal of educational Psychology 66.5 (1974): 688.

---

> > ### Comment · Reviewer_yvJa · 2023-08-20
> >
> > Thanks for the additional information and thoughtful responses. I am increasing my rating.

---

> > > ### Author Response · Authors · 2023-08-21
> > >
> > > We sincerely thank you for your constructive feedback and appreciate your acknowledgment of our attempts to address the concerns raised.

---

### Official Review · Reviewer_t8vT · 2023-07-06

**Soundness:** 3 good
**Presentation:** 4 excellent
**Contribution:** 3 good
**Rating:** 6
**Confidence:** 3

**Summary:**

This paper proposes a new estimator for the causal error, that achieves lower variance than previous approaches.
The estimator consists in the difference between a IPTW-like estimator using the causal model and a direct IPTW causal effect estimator.
The paper shows that under clear assumptions, this estimator results in lower variance than a naive estimator, which only uses a typical IPTW. The authors further test this estimator empirically on a wide range of setups that both satisfy and potentially violate their assumptions.

**Strengths:**

This paper propose a simple yet effective way to reduce the variance of the causal error estimation.

The assumptions under which the main result hold are very clearly stated and discussed.

The authors have extensively tested their approach. In particular, they have tested both on simulated data that satisfy their assumptions as well as on data that potentially violate them, to stress test the method.

**Weaknesses:**

The main theoretical comparison of this paper seems to be the naive IPTW estimator. As the authors state in the related works section, other estimators have been proposed in the literature to reduce the variance of the causal effect estimator. How does this estimator compare to those theoretically ? And can you leverage some the improvements of IPTW in your method too ?

I think the whole goal of the paper would deserve more clarity. For instance, the overall goal is usually to estimate the true causal effect rather the causal error. Having a good estimator of the causal effect would directly result in low causal error. I believe this would deserve some more motivation / details in the text.

Furthemore, I would encourage the author to make the problem setup more clear. In my opinion, it is not fully transparent from the paper if the model is trained on a different set of samples than the ones used for the IPTW estimator. I believe this is the case but this should be state more clearly.

**Questions:**

Same as above:

The main theoretical comparison of this paper seems to be the naive IPTW estimator. As the authors state in the related works section, other estimators have been proposed in the literature to reduce the variance of the causal effect estimator. How does this estimator compare to those theoretically ? And can you leverage some the improvements of IPTW in your method too ?

I think the whole goal of the paper would deserve more clarity. For instance, the overall goal is usually to estimate the true causal effect rather the causal error. Having a good estimator of the causal effect would directly result in low causal error. I believe this would deserve some more motivation / details in the text.

Furthemore, I would encourage the author to make the problem setup more clear. In my opinion, it is not fully transparent from the paper if the model is trained on a different set of samples than the ones used for the IPTW estimator. I believe this is the case but this should be state more clearly.

**Limitations:**

The limitations and assumptions of this work were carefully adressed.

---

> ### Author Rebuttal · Authors · 2023-08-09
>
> Thank you very much for your positive opinions for our work. We are pleased to see that you appreciate our efforts in technical soundness, presentation, as well as evaluations.
>
> > The main theoretical comparison of this paper seems to be the naive IPTW estimator. As the authors state in the related works section, other estimator s have been proposed in the literature to reduce the variance of the causal effect estimator. How does this estimator compare to those theoretically ? And can you leverage some the improvements of IPTW in your method too ?
>
> Thanks for your question. First, due to the variety of existing estimators, we cannot assert the theoretical advantages for each of them. However, we believe our results can be extended to the case of any extensions of self-normalized IPW, such as the Adaptive normalization for IPW estimation [1]. We will add discussions regarding this in the revision. Moreover, we would like to enhance that our paper focus on improving evaluation of causal inference models, rather than improving causal effect estimations itself.
>
> > Having a good estimator of the causal effect would directly result in low causal error. I believe this would deserve some more motivation / details in the text.
>
> We appreciate your suggestion and will provide better motivation in our revision. We would like to enhance that we did evaluate our method against other IPW variance reduction methods that are designed to obtain better causal effects; and our results indeed show that our method can still be very valuable when other effect-oriented estimators are not working well.
>
> Our method aims to offer a new, easy-to-use tool for causal practitioners, complementing other existing tools. By focusing on causal error estimation, we can address specific challenges in evaluating causal models.
>
> > Furthemore, I would encourage the author to make the problem setup more clear. In my opinion, it is not fully transparent from the paper if the model is trained on a different set of samples than the ones used for the IPTW estimator. I believe this is the case but this should be state more clearly.
>
> Thank you for pointing this out. We will make this clearer in our revision. As illustrated in Figure 1, there are two data sources: an observational data source for training the model and an interventional data source obtained through a designed oracle treatment assignment mechanism P(T=1|X) for validating the model. The IPW estimator is used to estimate the ground truth effect based on the interventional data, leveraging the known propensity scores P(T=1|X).
>
>
> **Reference**
>
> [1] Khan, Samir, and Johan Ugander. "Adaptive normalization for IPW estimation." Journal of Causal Inference 11.1 (2023): 20220019.

---

> ### Author Response · Authors · 2023-08-21
>
> Thank you again for your review, and we would like to ascertain if our rebuttal has adequately resolved the concerns you raised. We continue to welcome any supplementary observations or clarification to bolster our work.

---

### Official Review · Reviewer_zDTG · 2023-07-09

**Soundness:** 2 fair
**Presentation:** 2 fair
**Contribution:** 3 good
**Rating:** 5
**Confidence:** 3

**Summary:**

This paper constructs a new estimator for IPW evaluation by comparing the IPW estimator applied on the model-predicted treatments versus the observed treatments. The paper presents a theoretical result that this estimator has lower variance than the naive one and aims to demonstrate this via empirical experiments.

**Strengths:**

1. The structure of the paper and most of the writing are very good.

2. The theoretical result is important.

3. The empirical results seem to show that the estimator is better than the naive one.

**Weaknesses:**

1. The paper could benefit from more clarity in the writing. For example, in line 124, it would be great if there can be some intuition or example on when P(T=1|X) is skewed and what that means (is it overfitting or mis-specified)? Furthermore, it would be great if there was an example of a commonly used model and how it satisfies Assumption A, and why this assumption is not a strong one.

2. While the theoretical result seems important, the supplementary file with the proof is missing, which makes it impossible to review.

3. Similarly, the empirical experiments, while extensive, rely on datasets and details that are supposed to be in the supplementary file, but that file is missing. Therefore, it is unfortunately impossible to review the setting in detail.

**Questions:**

1. Could the authors briefly describe the derivation of the IPW estimator (the unlabeled equation between eq. 4 and 5). I have not seen it in this form before.

2. How do the authors explain the large spike in variance at \sigma_\beta = 10 in Figure 2 for the baseline?

**Limitations:**

The main limitation is the lack of the Appendix which contains significant details regarding the contributions of this paper. The authors should also discuss potential limitations of their theoretical assumptions.

---

> ### Author Rebuttal · Authors · 2023-08-09
>
> Thank you for your encouraging review and valuable suggestions to improve.
>
> > The paper could benefit from more clarity in the writing. For example, in line 124, it would be great if there can be some intuition or example on when P(T=1|X) is skewed and what that means (is it overfitting or misspecified)?
>
> We appreciate your suggestion and will clarify this point in our revision. In our paper, P(T=1|X) is a given oracle distribution that is known and designed by the experiment designer to evaluate the causal predictions of another causal model; and the goal of such experiment is to evaluate the predictions of a existing causal model.
>
> The skewness of P(T=1|X) is not due to overfitting or misspecification; rather, it often results from practical constraints such as ethical, legal, or financial considerations. For example, in business scenarios, any experimental interventions could have financial impact in the real world, leading to a trade-off between utility and test power. A more skewed policy might be preferred to minimize the negative impact of random assignment, at the cost of higher effect estimation variance. Our proposed method aims to address this challenge by providing a more reliable evaluation in such scenarios.
>
>
>
> > Furthermore, it would be great if there was an example of a commonly used model and how it satisfies Assumption A, and why this assumption is not a strong one.
>
> Thanks for your suggestion, relevant results are provided in the attached pdf in our general reply. We demonstrate with different models including linear DML, kernel DML, causal random forest, and variations of doubly robust algorithms (linear, forest, orthogonal), this assumption holds in practice naturally.
>
> > While the theoretical result seems important, the supplementary file with the proof is missing, which makes it impossible to review.
>
> We apologize for the omission of the appendix in our submission. This was an unintentional mistake, and we appreciate your understanding. Please refer to the proof provided in our response to Reviewer r45Q to review our theoretical results.
>
> We hope that our revised responses address your concerns more effectively. Please let us know if there are any further questions or suggestions.
>
> > Could the authors briefly describe the derivation of the IPW estimator (the unlabeled equation between eq. 4 and 5). I have not seen it in this form before.
>
> This is just a vectorized form of the usual IPW definition. Note that
>
> $
> \hat{\delta}^{IPW}(\mathcal{T})  = \frac{1}{N} \sum_{i\in \mathcal{B}} \frac{\mathbf{Y}^{T=1}(i)}{p_i} - \frac{1}{N} \sum_{j\in \mathcal{D} \setminus \mathcal{B}} \frac{\mathbf{Y}^{T=1}(j)}{(1 - p_j)} $
>
> $ = \frac{1}{N} \sum_{i\in \mathcal{B}} \frac{\mathbf{Y}^{T=1}(i)}{p_i} - \frac{1}{N} \sum_{j\in \mathcal{D} \setminus \mathcal{B}} \frac{\mathbf{Y}^{T=1}(j)/p_j}{(1/p_j - 1)} $
>
> $ = \frac{1}{N}<\mathbf{Y}^{T=1}(\mathcal{B}), \mathbf{w}(\mathcal{B})> - \frac{1}{N}<\mathbf{Y}^{T=0}(\mathcal{D} \setminus \mathcal{B}), \frac{\mathbf{w}(\mathcal{D} \setminus \mathcal{B})}{\mathbf{w}(\mathcal{D} \setminus \mathcal{B}) - 1}> .
> $
>
>
> > How do the authors explain the large spike in variance at \sigma_\beta = 10 in Figure 2 for the baseline?
>
> The spike of variance is expected since when $\sigma_\beta$ increases, the imbalance of treatment assignment would also increase.  Sometimes the variance might go down again at $ \sigma^2_\beta = 20$, but this almost only happens for the combination of certain baselines (e.g., naive IPW), certain $\sigma_\nu$ values (e.g., very small) and certain datasets (e.g., csuite_3 dataset). This might be due to that when $\sigma_\nu$ is very small, the model's bias will also be small, resulting in a causal error that has a much smaller scale, hence the smaller variance.

---

> > ### Comment · Reviewer_zDTG · 2023-08-20
> >
> > I appreciate the authors' response, I believe it addresses my questions and concerns. I am adjusting my score accordingly.

---

> > > ### Author Response · Authors · 2023-08-21
> > >
> > > Thank you for your valuable input, and we're grateful for the recognition of our efforts in tackling the issues brought up.

---

### Official Review · Reviewer_r45Q · 2023-07-24

**Soundness:** 3 good
**Presentation:** 3 good
**Contribution:** 3 good
**Rating:** 7
**Confidence:** 3

**Summary:**

This work aims to evaluate the fidelity of causal models in estimating true treatment effects across different treatments. The golden approach involves comparing treatment effects derived from the target causal model and those obtained from Randomized Controlled Trials (RCT). Practical, time, cost, and ethical constraints often necessitate replacing the RCT estimate with non-RCT methods such as Inverse Probability Weighting (IPW). However, IPW may lead to unbounded variance due to imbalanced propensity scores. To address this, the authors introduce a procedure that applies the IPW estimator to both the model and the actual effects. This aligns the estimated treatment effects, thus offsetting their estimation errors, and results in a lower variance causal error estimate. Under the two stated assumptions, the authors show that the variance of the causal error estimated from their approach, namely pairs estimator, is upper bounded by variance of the the causal error estimated from the naive estimator. In their experiments, the authors compared their approach with the naive estimator, RCT estimator, and existing state-of-the-art variance reduction estimators such as the self-normalized estimator and the LW IPW estimator. They carried out these comparisons on three synthetic datasets, under various non-RCT scenarios, which included different treatment assignment units across sub-populations and varying degrees of propensity score imbalance. The results demonstrated that their approach consistently produced low estimation errors, often on par with those from the RCT estimator. Moreover, they also evaluated the performance of their approach when the existing machine learning-based causal models are used in treatment effect estimation. Still, the pairs estimator yield low estimation errors and yielded results comparable to those from the RCT estimator.

**Strengths:**

1. The author proposes a simple yet powerful procedure for estimating causal error without modifying IPW, which might lead to other complexity, such as parameter tuning, and bias introduction. Their experiments demonstrate their approach has the capability in estimating true causal error faithfully under many existing non-RCT scenarios that are often encountered in practice.

2. The problem statement, formulation, and illustration are clearly stated and well structured, allowing readers to follow easily.

**Weaknesses:**

Despite their approach being supported by the theoretical results and extensive experiments presented, the authors have not provided an appendix. The thorough justification of their theoretical results and experimental details could only be further substantiated with access to this supplementary material.

**Questions:**

1. Regarding Figure 2, can you clarify why the variance of the Linear-modified estimator and Normalized IPW initially increase and subsequently decrease as the imbalance degree increases? Wouldn't these variance reduction methods yield improved results when the degree of imbalance is less pronounced?

2. In assumption A, what is $b_i$?

3. Given that your theoretical findings strongly depend on Assumption A, the compliance of the learned causal model's counterfactual predictions with this assumption becomes a key aspect of your experimental inquiry. Could you please provide the appendix and discuss these results in more detail?

####################################################################################

[08/19/ 2023] Reviewer r45Q: The experiment validation on Assumption A and the proof for Proposition 1 are provided, and hence I adjust my review accordingly.

####################################################################################

**Limitations:**

To my knowledge, this work does not have potential negative societal impacts. However, the authors did not provide a section with these discussions.

---

> ### Author Rebuttal · Authors · 2023-08-09
>
> Thank you so much for your constructive and positive feedback to our paper. We would address your comments below.
>
> > Despite their approach being supported by the theoretical results and extensive...the authors have not provided an appendix.
>
> We apologize for the missing appendix, this is due to our unpurposive mistakes. We will add it in the revision. Regarding the theoretical results, they are appended at the end of this response since the proof is not complicated.
>
> > The compliance of the learned causal model's counterfactual predictions with this assumption becomes a key aspect of your experimental inquiry. Could you please provide the appendix?
>
> Yes, please see the pdf file our general reply.
>
> > In assumption A, what is $b_i$?
>
> $b_i$ is given in the paragraph below Eq (4), which is the Bernoulli random variable of treatment assignment.
>
> **Proof of Prop. 1**:
>
> First, it is straightforward to verify that under **Assumption A**, $\hat{\Delta}^{Pairs}(\mathcal{M}, \mathcal{T})$ and $\hat{\Delta}(\mathcal{M}, \mathcal{T})$ can be decomposed as
>
> $ \hat{\Delta}^{Pairs}(\mathcal{M}, \mathcal{T}) =  \hat{\delta}_\mathcal{M} - \hat{\delta}  + g(\mathbf{\nu}, \mathcal{B}) $
>
> $ \hat{\Delta}(\mathcal{M}, \mathcal{T}) = \hat{\delta}_\mathcal{M} - \hat{\delta}   - f(\mathcal{B})$
>
> Where:
>
>  $f(\mathcal{B}) = \frac{1}{N} <\mathbf{Y}^{T=1} (\mathcal{B}), \mathbf{w}(\mathcal{B}) - 1 >   + \frac{1}{N} <\mathbf{Y}^{T=0}(\mathcal{B}), \mathbf{1}>  $  \
> $- \frac{1}{N}<\mathbf{Y}^{T=0}(\mathcal{D} \setminus \mathcal{B}), \frac{1}{\mathbf{w}(\mathcal{D} \setminus \mathcal{B}) - 1} > - \frac{1}{N}<\mathbf{Y}^{T=1}(\mathcal{D} \setminus \mathcal{B}), \mathbf{1}> $,
>
> And:
>
> $g(\mathbf{\nu}, \mathcal{B})  = \frac{1}{N}<\mathbf{\nu}(\mathcal{B}) * \mathbf{v}(\mathbf{Y}^{T=1} (\mathcal{B})), \mathbf{w}(\mathcal{B}) - 1 >    \frac{1}{N}<\mathbf{\nu}(\mathcal{B}) *\mathbf{v}(\mathbf{Y}^{T=0}(\mathcal{B})), \mathbf{1}> $ \
> $ - \frac{1}{N}<\mathbf{\nu}(\mathcal{D} \setminus \mathcal{B}) *\mathbf{v}(\mathbf{Y}^{T=0}(\mathcal{D} \setminus \mathcal{B})), \frac{1}{\mathbf{w}(\mathcal{D} \setminus \mathcal{B}) - 1} > - \frac{1}{N}<\mathbf{\nu}(\mathcal{D} \setminus \mathcal{B})*\mathbf{v}(\mathbf{Y}^{T=1}(\mathcal{D} \setminus \mathcal{B})), \mathbf{1}>  $ .
>
> Their estimation error can then be expressed as
>
> $ e(\hat{\Delta}^{Pairs}(\mathcal{M}, \mathcal{T}))  =  \hat{\delta}_\mathcal{M} - \hat{\delta}  + g(\mathbf{\nu}, \mathcal{B}) - \Delta(\mathcal{M}) $
>
> $ e(\hat{\Delta}(\mathcal{M}, \mathcal{T})) =  \hat{\delta}_\mathcal{M} - \hat{\delta}   - f(\mathcal{B}) - \Delta(\mathcal{M}) $
>
>
> According to delta method, both $\sqrt{N} e(\hat{\Delta}^{Pairs})$ and $\sqrt{N} e(\hat{\Delta})$ are asymptotically normal with zero mean under Assumption B.
>
> Note that $g(\mathbf{\nu}, \mathcal{B})$ can be rewritten as
>
>    $  g(\mathbf{\nu}, \mathcal{B})  = \frac{1}{N}<\pmb{\mathbf{\nu}} *\mathbf{b}*\mathbf{v}(\mathbf{Y}^{T=1}), \mathbf{w} - 1 >   + \frac{1}{N}<\mathbf{\nu} *\mathbf{b} *\mathbf{v}(\mathbf{Y}^{T=0}), \mathbf{1}> $ \
>    $ - \frac{1}{N}<\mathbf{\nu} *(1-\mathbf{b}) *\mathbf{v}(\mathbf{Y}^{T=0}), \frac{1}{\mathbf{w} - 1} > - \frac{1}{N}<\mathbf{\nu}*(1-\mathbf{b}) *\mathbf{v}(\mathbf{Y}^{T=1}), \mathbf{1}> $
>
> where $b_i$ is the Bernoulli variable with $P(b_i=1) = p_i$, and $b_i = 1$ if $i \in \mathcal{B}$.
>
> Note also that $\nu$ is independent from $(Y^{T}(i), b_i)$ and has zero mean and variance $\sigma$, we have
>
> $ Cov(Y^{T=t_a}(i), \nu_i b_i V_i(Y^{T=t_b}(i)) ) = E(\nu_i b_i) Cov(Y^{T=t_a}(i), V_i(Y^{T=t_b}(i)) ) = 0$
>
> holds for all $i$ and all treatments $t_a$ and $t_b$. Similarly, we have $ Cov(Y_\mathcal{M}^{T=t_a}(i), \nu_i b_i V_i(Y^{T=t_b}(i)) ) =0$.
>
> Therefore, it is not hard to show that $ Cov( g(\mathbf{\nu}, \mathcal{B}), \hat{\delta}_\mathcal{M}) = 0 $, and $Cov( g(\mathbf{\nu}, \mathcal{B}), \hat{\delta} ) = 0$,
>
> which implies $Cov( g(\mathbf{\nu}, \mathcal{B}), \hat{\delta}_\mathcal{M} - \hat{\delta} ) = 0 $. Thus, we have:
>
> $
>     Var[ \sqrt{N} e(\hat{\Delta}^{Pairs}(\mathcal{M}, \mathcal{T})) ]  =  Var[ \sqrt{N} (\hat{\delta}_\mathcal{M} - \hat{\delta})]  + Var[\sqrt{N}g(\mathbf{\nu}, \mathcal{B})] $
>
> $ = Var[Y_\mathcal{M}^{T=1} - Y_\mathcal{M}^{T=0} + Y^{T=1} - Y^{T=0}] +  Var[\sqrt{N}g(\mathbf{\nu}, \mathcal{B})]\\
>      Var[\sqrt{N}e(\hat{\Delta}(\mathcal{M}, \mathcal{T}))]  $
>
> $= Var[\sqrt{N}( \hat{\delta}_\mathcal{M} - \hat{\delta})]  + Var[ \sqrt{N(}f(\mathcal{B}))] $
>
>   $ = Var[Y_\mathcal{M}^{T=1} - Y_\mathcal{M}^{T=0} + Y^{T=1} - Y^{T=0}] + Var[ \sqrt{N(}f(\mathcal{B}) )] $.
>
>
> Since $\nu$ has zero mean and variance $\sigma^2_\nu$ and is independent of $(Y_i^{T}, b_i)$ as in Assumption A, this expression be further simplified as, according to the rules of variance of the product of independent variables:
>
> $    Var[g(\mathbf{\nu}, \mathcal{B})]  = \frac{1}{N^2}<\sigma^2_\nu *\mathbf{p}*E(\mathbf{v}(\mathbf{Y}^{T=1})^2), (\mathbf{w} - 1)^2 >   + \frac{1}{N^2}<\sigma^2_\nu *\mathbf{p} * E(\mathbf{v}(\mathbf{Y}^{T=0})^2), \mathbf{1}>  $
>
> $     + \frac{1}{N^2}<\sigma^2_\nu *(1-\mathbf{p}) *E(\mathbf{v}(\mathbf{Y}^{T=0})^2), \frac{1}{\mathbf{w} - 1} > + \frac{1}{N^2}<\sigma^2_\nu*(1-\mathbf{p}) * E(\mathbf{v}(\mathbf{Y}^{T=1})^2), \mathbf{1}> $
>
> $     = \frac{\sigma^2_{\nu}}{N^2}[ <\mathbf{p}(\mathbf{w}-1)^2 + (\mathbf{1}-\mathbf{p}), E((\mathbf{v}(\mathbf{Y}^{T=1})^2)> +  <\frac{1-\mathbf{p}}{\mathbf{w} - 1} + (\mathbf{p}), E\mathbf{v}(\mathbf{Y}^{T=0})^2)>] $
>
> $     < \frac{1}{N^2}[ <\mathbf{p}(\mathbf{w}-1)^2 + (\mathbf{1}-\mathbf{p}), E((\mathbf{Y}^{T=1})^2)> +  <\frac{1-\mathbf{p}}{\mathbf{w} - 1} + (\mathbf{p}), E((\mathbf{Y}^{T=0})^2)>]     = Var[f(\mathcal{B})] $.
>
> Where the third equality is due to the fact that $ \sigma^2_\nu E [(V_i Y^{T=t}(i))^2] < E [Y^{T=t}(i)^2]$. Therefore, we finally conclude that the variances of the error estimators will satisfy
> $ Var[ \sqrt{N} e(\hat{\Delta}^{Pairs}(\mathcal{M}, \mathcal{T})) ] < Var[\sqrt{N}e(\hat{\Delta}(\mathcal{M}, \mathcal{T}))]$.
>
> **QED**

---

> > ### Comment · Reviewer_r45Q · 2023-08-19
> >
> > The experiment validation on Assumption A and the proof for Proposition 1 are provided, and hence I adjust my review accordingly.

---

> > > ### Author Response · Authors · 2023-08-21
> > >
> > > Thank you again for your feedback and we appreciate the recognition of our efforts in addressing the concerns raised.

---

### Official Review · Reviewer_PdWH · 2023-07-30

**Soundness:** 3 good
**Presentation:** 3 good
**Contribution:** 3 good
**Rating:** 6
**Confidence:** 3

**Summary:**

The paper considers the estimation of causal error using the IPW estimator in conditional randomized experiments. Given that the allocation probabilities are readily available in these types of experiments, IPW estimators are often used. The authors propose to use the same IP weights for both the causal prediction, and the the estimator of the ground truth, and show that this so-called pair estimation approach reduces the variance in the distance between the causal prediction and ground truth. They provide theoretical justifications for their approach in terms of variance reduction, and illustrate this on synthetic data sets.

**Strengths:**

The idea of using the same IP weights for both the causal prediction and ground truth estimation is simple yet effective.

**Weaknesses:**

There is no evaluation of real data sets.

Often the IPW is bad when the ps is unknown and the model is misspecified. When the ps is known by design, this seems to be less of a concern, and can/should probably be addressed by a better design.

Often the IP weights are used in conjunction with an OR model to construct a DR estimator. In fact, there is really no good reasons to use the naive IPW estimator considered by the authors. I doubt the proposed method may also be useful for improving the DR estimator, but not as dramatic. The authors should probably have done this.

**Questions:**

The authors refer to the phenomenon that the oracle IPW may not be as efficient as the one using a correctly specified model as one of the limitation of the IPW estimator. Why is this a limitation?

**Limitations:**

The author’s use of non-RCT data is very misleading; this is actually often called the conditional randomized studies (e.g. Imbens and Rubins’ book), and is one common type of randomized studies. A trial does not need to have the same coin for everyone!

The authors mention that their approach is different from the modern approaches that try to stablize the IP weights. It is unclear to me if one already use these stablization methods, then whether the authors’ method is still useful.

---

> ### Author Rebuttal · Authors · 2023-08-09
>
> Thank you for the positive feedback and suggestions. Below, we respond to each of your comments.
>
>
> > Often the IP weights are used in conjunction with an OR model to construct a DR estimator. In fact, there is really no good reasons to use the naive IPW estimator considered by the authors.
>
> We acknowledge the effectiveness of DR estimators in causal inference settings. However, our paper focuses on **model evaluation in real-world scenarios** where IP weights are known by design, and model misspecification is less of a concern. As such, the introduction of the DR estimator might not be necessary or optimal. Instead, the high variance issue of IPW is more of a concern, which cannot be mitigated by D. We compared our method with existing IPW variance reduction techniques, such as self-normalized/renormalized [1, 2] and linearly modified (LM) IPW estimators[3], demonstrating the effectiveness of our approach.
>
> > Often the IPW is bad when the ps is unknown and the model is misspecified. When the ps is known by design, this seems to be less of a concern, and can/should probably be addressed by a better design.
>
> While it is true that known propensity scores alleviate some concerns, IPW might still suffer from high variance, leading to poor estimates. Addressing this issue through a different design is often infeasible due to practical limitations, such as financial concerns, or ethical constraints. Our method provides a solution that works within these constraints, helping to improve causal model evaluation even when propensity scores are known by design.
>
>
> > There is no evaluation of real data sets.
>
> We recognize the absence of real dataset evaluations as a weakness of our paper. However, it is very difficult to find such real-world datasets that satisfies: 1) the non RCT treatment assignment has known oracle propensity scores; 2) the interventional outcomes are not generated in a semi-synthetic manner. Otherwise, these “real” datasets would be no different from our experiments. Our method was inspired by a real-world problem we encountered, and we achieved good results in our actual application, motivating us to share the method that we discovered. Due to confidentiality reasons, we cannot share these proprietary results and data publicly. That said, we believe our simulation studies sufficiently demonstrate the potential of our method in various scenarios.
>
>
> > The author’s use of non-RCT data is very misleading; this is actually often called the conditional randomized studies (e.g. Imbens and Rubins’ book), and is one common type of randomized studies. A trial does not need to have the same coin for everyone!
>
> Thank you for the suggestion. We recognize that there is different convention of naming such as “non-RCT” used in [4] but CRS in other literatures.  In revision, we will use the suggested terms and mention different conventions in the literature.
>
> > It is unclear to me if one already use these stablization methods, then whether the authors’ method is still useful.
>
> In our experiments, we have indeed compared the performance of our method versus some stabilization methods such as renormalization and linear corrections; and our results showed that our method will still be helpful even if these methods are already available.
>
> **Reference**
>
> [1] Imbens, Guido W. "Nonparametric estimation of average treatment effects under exogeneity: A review." Review of Economics and statistics 86.1 (2004): 4-29.
>
> [2] Lunceford, Jared K., and Marie Davidian. "Stratification and weighting via the propensity score in estimation of causal treatment effects: a comparative study." Statistics in medicine 23.19 (2004): 2937-2960.
>
> [3] Zhou, Kangjie, and Jinzhu Jia. "Variance Reduction for Causal Inference." arXiv preprint arXiv:2109.05150 (2021).
>
> [4] Rubin, Donald B. "Estimating causal effects of treatments in randomized and nonrandomized studies." Journal of educational Psychology 66.5 (1974): 688.

---

> > ### Author Response · Authors · 2023-08-21
> >
> > We would like to thank you again for spending time carefully evaluating our submission and providing valuable feedback. We would appreciate if you could let us know if our rebuttal has addressed your concerns, thank you.

---

### Author Rebuttal · Authors · 2023-08-09

We thank the reviewers for your encouraging review and valuable suggestions to improve. We acknowledge that the reviewers highlighted the **effectiveness and simplicity of our approach (PdWH, r45Q,  t8vT, yvJa), novelty or soundness of our results (PdWH, zDTG,  t8vT, yvJa), extensiveness of experiments (r45Q, r45Q,  t8vT), and clarity of presentation (PdWH, r45Q, zDTG,  t8vT, yvJa)**.

- An essential aspect of our work that we would like to emphasize is its focus on **model evaluation in real-world testing scenarios**. Our novel method for low-variance estimation of causal error facilitates reliable evaluation of causal inference models through **oracle conditional/non-randomized trial**. This is particularly relevant in situations where the oracle design cannot be improved due to real-world constraints or interests. Evaluating causal models in these settings is crucial to ensure their validity and applicability across various domains. Our approach is simple yet powerful, complementing existing tools in the causal inference toolbox, effectively addressing specific challenges faced by practitioners in a wide range of real-world applications.

- Moreover, we have **addressed the main concerns raised by the reviewers**, including:

    - as requested, providing **experiments to validate Assumption A**. In the attached pdf file, using an numerical example we have demonstrated that Assumption A holds in practice for various commonly used models, including linear DML, kernel DML, causal random forest, and variations of doubly robust algorithms (linear, forest, orthogonal). **This indicates that Assumption A is not overly restrictive and can be satisfied by a wide range of causal models that are popular in practical applications**.

    - as requested, providing **theoretical proofs** to the main result,  which can be found in our response to **r45Q**.

We believe that our responses effectively address the concerns raised by the reviewers and hope that they find our work more compelling and valuable. We are grateful for the reviewers' insightful feedback and look forward to incorporating their suggestions to further improve the quality of our paper.

---

### Decision · Program_Chairs · 2023-09-21

**Decision:**

Accept (poster)

**Comment:**

The paper presents a novel estimator for evaluating causal models in the context of non-randomized experiments. This approach aims to reduce the variance in causal error estimation. The authors provide theoretical justifications for the method and conduct empirical experiments to demonstrate its effectiveness. There were some questions and concerns raised by reviewers including the lack of discussion about failure modes or the practical scenarios where the required model of treatment effect and non-randomized data for evaluation are both available. The paper would benefit from addressing these comments.